# Evaluation of the Rheological Properties of Virgin and Aged Asphalt Blends

**DOI:** 10.3390/polym14173623

**Published:** 2022-09-01

**Authors:** Tao Liu, Weidang Duan, Jialin Zhang, Qiuping Li, Jian Xu, Jie Wang, Yongchun Qin, Rong Chang

**Affiliations:** 1Jiangxi Provincial Communications Investment Group Co., Ltd., Nanchang 330025, China; 2Research Institute of Highway, Ministry of Transport, Beijing 100088, China

**Keywords:** road engineering, reclaimed asphalt binder, rheological properties, SHRP test, variance analysis

## Abstract

To evaluate the effects of the source and admixture of aged asphalt on the rheological properties of reclaimed asphalt binders, the relative viscosity (Δ*η*), relative rutting factor (ΔG*/sinδ), and relative fatigue factor (ΔG*sinδ) were selected as evaluation indicators based on the Strategic Highway Research Program (SHRP) tests to characterize the rheological properties of a reclaimed asphalt binder under medium- and high-temperature conditions. The results of the study showed that the viscosity, rutting factor, and fatigue factor of the reclaimed asphalt binder increased with the addition of aged asphalt; however, the effect of the source and admixture of aged asphalt could not be assessed. The relative viscosity, relative rutting factor, and relative fatigue factor are sensitive to the source, admixture, temperature, and aging conditions, which shows the superiority of these indicators. Moreover, the relative viscosity and relative rutting factor decreased linearly with increasing temperature under high-temperature conditions, while the relative fatigue factor increased linearly with increasing temperature under medium-temperature conditions. In addition, the linear trends of the three indicators were independent of the source and admixture of aged asphalt. These results indicate that the evaluation method used in this study can be used to assess the effects of virgin asphalt and aged asphalt on the rheological properties of reclaimed asphalt binders, and has the potential for application. The viscosity of recycled asphalt increases, and the rutting factor and fatigue factor both increase. The high-temperature stability of reclaimed asphalt is improved, and the fatigue crack resistance is weakened.

## 1. Introduction

Reclaimed asphalt pavement (RAP) contains large amounts of aggregate and asphalt, which are potentially usable resources [1,2,3,4]. The recycling of RAP contributes to reductions in rock mining and aggregate production, with significant economic and environmental benefits [5,6]. Studies [2,7] have shown that the application of reclaimed asphalt mixtures in pavement subgrade construction can reduce greenhouse gas emissions by 20%, energy consumption by 16%, hazardous waste by 11% (RAP may leach toxic substances such as polycyclic aromatic hydrocarbons (PAHs) in the presence of rainwater in long-term stockpiles), and whole-life costs by 21%.

Asphalt is a viscoelastic material with excellent rheological properties [8,9]. After the blending of virgin and aged asphalt, the reclaimed asphalt binder becomes viscous and produces a large variation in rheology compared to the virgin asphalt [10,11,12]. RAP comes from a wide range of sources and has a very complex composition. Differences in the asphalt, aggregates, oil-to-rock ratio, gradation, and even admixtures may lead to significant differences in the rheology of the reclaimed asphalt binder or in the road performance of the recycled mix [11,13]. Numerous studies and engineering experiences have shown that pavements paved with poorly rheological asphalt are prone to high-temperature rutting, fatigue cracking, and other diseases [14,15]. In addition, the differences in the rheology of reclaimed asphalt binder will also lead to differences in the construction temperature of the mixture, which in turn will cause differences in the fuel required in the plant mix and greenhouse gas emissions [13,16].

Therefore, to optimize the design of reclaimed asphalt mixtures, reduce the reclaimed pavement issues, and prolong the service life, the rheological properties of reclaimed asphalt binders need to be evaluated effectively. At present, there are few studies on this subject, and the effects of different sources and admixtures of aged asphalt on the rheology of reclaimed asphalt binders under different temperatures and aging conditions have not been fully considered [10,17,18]. This study selects the Strategic Highway Research Program (SHRP) asphalt test method to quantitatively evaluate the effects of the source and admixture of aged asphalt on the high-temperature rheology of reclaimed asphalt binders using the relative viscosity, relative rutting factor, and relative fatigue factor.

## 2. Materials and Method

### 2.1. Materials

This study uses the Abson method (ASTM D 1856) to reclaim asphalt from asphalt mixtures. The asphalt mixture was extracted with a distillation device, and then the solvent in the extraction liquid was removed. The recovered asphalt samples were denoted by A, B, and C. The performance indexes are shown in Table 1. Asphalt with PG 90 was used for the virgin asphalt and denoted by N. The performance indicators of the virgin asphalt are shown in Table 2.

The aged and virgin asphalts were mixed uniformly at 135 °C, and the admixtures of the aged asphalts were 15% and 30%, respectively. It is worth mentioning that the admixture refers to the ratio of the mass of aged asphalt to the total mass of reclaimed asphalt binder. The thin-film oven test (TFOT) and pressure aging vessel (PAV) were used to simulate short-term aging and long-term aging, respectively. For the sake of simplicity, the following abbreviations are used in this study: W for unaged, D for short-term aging, and C for long-term aging. For example, A15W represents virgin asphalt mixed with 15% of aged asphalt A, which was not aged; B30D represents virgin asphalt mixed with 30% of aged asphalt B, which underwent short-term aging.

### 2.2. Asphalt Viscosity Evaluation Test

The Brookfield rotational viscosity test (T0625) was used to investigate the effects of the source and admixture of aged asphalt on the high-temperature rheology of reclaimed asphalt binders. The viscosities of virgin asphalt and reclaimed asphalt binders were tested under unaged and short-term aging conditions, respectively. Considering the temperature range of 130–160 °C for asphalt production and application, the test temperatures were set at 115 °C, 125 °C, 135 °C, 145 °C, 155 °C, and 165 °C.

### 2.3. Asphalt Viscoelasticity Evaluation Test

The rheological properties of virgin and reclaimed asphalt binders at high and medium temperatures were investigated using dynamic shear rheometer (DSR) tests (T0628). The test plate was a circular metal plate with a diameter of 25 mm. The test temperatures for the unaged and short-term aged asphalts were 52 °C, 58 °C, 64 °C, 70 °C, 76 °C, and 82 °C, and the test temperatures for long-term aged asphalt were 16 °C, 19 °C, 22 °C, 25 °C, 28 °C, and 31 °C. It should be noted that the test temperatures refer to the American Association of State Highway and Transportation Officials (AASHTO T315) regulations.

## 3. Results and Discussion

### 3.1. Asphalt Viscosity Evaluation Test Results and Analysis

The viscosities of the asphalts at each test temperature are listed in Table 3. Each asphalt sample was tested 3 times in parallel. The test results satisfy the allowable error of the repeatability test being 3.5% of the average value.

The viscosities of the virgin and reclaimed asphalt binders decreased gradually with increasing temperature, independent of whether they were aged or not. The viscosity of the reclaimed asphalt was significantly greater than that of the virgin asphalt, while the viscosity of the short-term aged asphalt was significantly greater than that of the virgin reclaimed asphalt.

To evaluate the influence of the source and admixture of the aged asphalt on the viscosity of the reclaimed asphalt, the relative viscosity Δ*_η_* was used for the evaluation, which mainly characterizes the influence of the relative viscosity of the reclaimed asphalt on its viscosity for every 1% increase in the content of aged asphalt, calculated as Equation (1):(1)$$\Delta_{\eta }=\frac{\eta_{mix.r}-\eta_{new.r}}{x}$$
where Δ*_η_* is the dimensionless viscosity of the reclaimed asphalt, *η_mix.r_* is the relative viscosity of the reclaimed asphalt, *η_mix.r_* = *η_mix_*/*η_new_*, *η_mix_* is the viscosity of the reclaimed asphalt (Pa·s), *η_new_* is the viscosity of the virgin asphalt (Pa·s), *η_new.r_* is the relative viscosity of the virgin asphalt, *η_new.r_* = 1, and *x* is the amount of aged asphalt blending.

The variance results for the Δ*_η_* and viscosity *η* values of the reclaimed asphalt samples at different test temperatures and under different aging conditions are listed in Table 4. Usually, the significance level α = 0.05.

As can be seen from Table 4, the statistical probability *p*-value of *η* is less than 0.05 only for the test temperature and aging conditions, indicating that there is no significant difference in the effects of virgin asphalt and aged asphalt on the viscosity of reclaimed asphalt based on the η index. For Δ*_η_*, the *p*-values for all four influencing factors are less than 0.05, indicating that assessing the viscoelasticity of reclaimed asphalt with Δ*_η_* can identify the differences in these four factors. Therefore, the high-temperature rheology of the reclaimed asphalt is better assessed using Δ*_η_* than the viscosity index.

The variation in relative viscosity Δ*_η_* of the reclaimed asphalt versus temperature is shown in Figure 1. As can be seen from Figure 1, the Δ*_η_* of the reclaimed asphalt gradually decreases with the increase in temperature, and after short-term aging the Δ*_η_* also gradually decreases. When the temperature is 135 °C, the Δ*_η_* values of *A*15*_W_*, *B*15*_W_*, and *C*15*_W_* are 2.46, 2.11, and 1.75, respectively, indicating that the relative viscosity increases by 2.46, 2.11, and 1.75 for each 1% increase in the content of aged asphalt under this test condition. However, under the same temperature conditions, the Δ*_η_* of B30_W_ is 2.63, which is not the same as that of B15_W_, indicating that the viscosity of the reclaimed asphalt produces inconsistent changes under different aged asphalt admixtures, even if the aged asphalt is the same. Moreover, the different reclaimed asphalts at different test temperatures and different aging conditions produced similar test results as described above.

The regression analysis results show that the reclaimed asphalt Δ*_η_* has a good linear relationship with the test temperature (T). The regression equations are all Δ*_η_* = a*T* + b (a and b are the fitting parameters), as shown in Equations (2) and (3).
(2)$$\Delta_{\eta }=\left\{\begin{array}{l} -0.0157T+4.6776,R^{2}=0.92,A15_{W}\\ -0.0174T+5.3464,R^{2}=0.97,A30_{W}\\ -0.0144T+4.1637,R^{2}=0.92,B15_{W}\\ -0.0134T+4.4118,R^{2}=0.87,B30_{W}\\ -0.0151T+3.8304,R^{2}=0.97,C15_{W}\\ -0.02T+4.8268,R^{2}=0.88,C30_{W} \end{array}\right.$$
(3)$$\Delta_{\eta }=\left\{\begin{array}{l} -0.0138T+4.2984,R^{2}=0.91,A15_{D}\\ -0.018T+5.209,R^{2}=0.89,A30_{D}\\ -0.0179T+4.4458,R^{2}=0.98,B15_{D}\\ -0.0186T+4.8137,R^{2}=0.96,B30_{D}\\ -0.0146T+3.6383,R^{2}=0.87,C15_{D}\\ -0.0212T+4.8106,R^{2}=0.98,C30_{D} \end{array}\right.$$

From Equations (2) and (3), it can be seen that the effects of virgin asphalt and aged asphalt on the high-temperature rheology of the reclaimed asphalt can be characterized by the slope and intercept in the linear relationship equation. For example, the slope of *B*30*_W_* is −0.013, which is 23% and 33% smaller than for *A*30*_W_* and *C*30*_W_*, respectively. In addition, the Δ*_η_* of the reclaimed asphalt is always linearly related to the temperature, independent of the source, the admixture of the aged asphalt, or whether it undergoes short-term aging.

### 3.2. Asphalt Rutting Factor Test Results and Analysis

The DSR test results of the asphalts under different test conditions are shown in Figure 2, Figure 3 and Figure 4.

The complex modulus G* can describe the ability of the asphalt to resist deformation, and the δ can reflect the proportional relationship between the elastic and viscous parts of the asphalt. Generally speaking, the larger δ is, the more viscous the asphalt is. From Figure 2, Figure 3 and Figure 4, it can be seen that as the temperature increases, G* decreases and δ increases, and the regularity is not related to the source of the aged asphalt, the admixture of the aged asphalt, or whether it has been aged. The G* of the reclaimed asphalt is greater than that of the virgin asphalt under each test temperature condition, and the δ of the reclaimed asphalt is less than that of the virgin asphalt, whereby the lower the temperature the more significant the result. In addition, the aged asphalt with the higher admixture content has a larger G* and smaller δ.

Asphalt under the long-term coupling effects of heat, oxygen, light, water, and load will experience serious aging, which will be manifested in the components as a decrease in aromatic content, an increase in asphalt content, and a macroscopic increase in hardness. From a viscoelastic point of view, the viscosity of asphalt decreases, the elasticity increases, and the asphalt changes from the sol–gel state to the gel state, which leads to a higher G* and lower δ. 

Adding a different proportion of aged asphalt to the virgin asphalt can improve the high-temperature performance of recycled asphalt; that is, the ability of the asphalt to resist high-temperature deformation. This is manifested as an increase in G* and a decrease in δ. According to the changes of G* and δ, it is considered that adding virgin asphalt to the aged asphalt can restore the rheological properties of the aged asphalt mixture.

### 3.3. Asphalt Rutting Factor Evaluation Test Results and Analysis

The road rutting is the irrecoverable deformation of asphalt pavement under the coupling effect of load and high temperature, which can be evaluated by using the rutting factor (G*/sinδ). The variation curves of the G*/sinδ with temperature for the virgin and reclaimed asphalts are shown in Figure 5.

As shown in Figure 5, the G*/sinδ values of the virgin and reclaimed asphalts gradually decreased with the increase in temperature. The G*/sinδ of the reclaimed asphalt was larger than that of the virgin asphalt. The nonlinear regression analysis showed that the G*/sinδ had a good exponential relationship with the temperature, and the correlation coefficients were all above 0.90.

In order to evaluate the effects of the source and admixture of the aged asphalt on the rutting factor of the reclaimed asphalt, the dimensionless rutting factor Δ_G*/sinδ_ was evaluated, and the calculation can be found in Equation (4):(4)$$\Delta_{G*/\sin\delta }=\frac{G*/\sin\delta_{mix.r}-G*/\sin\delta_{new.r}}{x}$$
where Δ_G*/sinδ_ is the dimensionless rutting factor of the reclaimed asphalt, G*/sinδ*_mix.r_* is the relative rutting factor of the reclaimed asphalt, G*/sinδ*_mix.r_* = (G*/sinδ*_mix_*)/(G*/sinδ*_new_*), G*/sinδ*_mix_* is the rutting factor of the reclaimed asphalt, G*/sinδ*_new_* is the rutting factor of the virgin asphalt, G*/sinδ*_new.r_* is the relative rutting factor of the virgin asphalt, G*/sinδ*_new.r_* = 1, and *x* is the amount of aged asphalt mixing.

The variance results for Δ_G*/sinδ_ and G*/sinδ for reclaimed asphalt at different test temperatures and under different aging conditions are shown in Table 5 with the significance level of *α* = 0.05.

As can be seen from Table 5, the statistical probability *p*-value of G*/sinδ is only less than 0.05 under one test temperature, indicating that using G*/sinδ as an indicator to evaluate the high-temperature stability of the reclaimed asphalt under different aging conditions is unable to distinguish the difference between the virgin asphalt and aged asphalt. The four *p*-values of Δ_G*/sinδ_ are less than 0.05, indicating that using Δ_G*/sinδ_ as an indicator to evaluate the high-temperature performance of the reclaimed asphalt under different temperature and aging conditions can distinguish the differences between virgin asphalt and aged asphalt. Therefore, it is more reasonable to use Δ_G*/sinδ_ as an indicator.

The variation in Δ_G*/sinδ_ versus temperature for the reclaimed asphalt is shown in Figure 6. It can be found that the Δ_G*/sinδ_ of the reclaimed asphalt decreases gradually with the increase in temperature. The Δ_G*/sinδ_ values for A30_W_, B30_W_, and C30_W_ at the test temperature of 58 °C were 16.0, 11.8, and 9.2, respectively, indicating that the relative rutting factors of the aged asphalt increased by 16.0, 11.8, and 9.2 for each 1% increase in the admixture of aged asphalt. Under the same temperature conditions, the relative rutting factor of A15_W_ was 14.9, which was not the same as that of A30_W_, indicating that every 1% increase in the admixture of aged asphalt produced inconsistent changes in the high-temperature performance of the reclaimed asphalt, even if the aged asphalt was from the same source. Different reclaimed asphalts at different test temperatures and different aging conditions will produce similar test results as above, which are similar to the relative viscosity test results.

The results of the regression analysis show that Δ_G*/sinδ_ has a good linear relationship with the test temperature (T). The regression equations are all Δ_G*/sinδ_ = a*T* + b (a and b are fitting parameters), as shown in Equations (5) and (6):(5)$$\Delta_{G*/\sin\delta }=\left\{\begin{array}{l} -0.5518T+44.855,R^{2}=0.92,A15_{W}\\ -0.4383T+40.44,R^{2}=0.92,A30_{W}\\ -0.4957T+39.797,R^{2}=0.92,B15_{W}\\ -0.3548T+31.908,R^{2}=0.91,B30_{W}\\ -0.3799T+29.819,R^{2}=0.83,C15_{W}\\ -0.2861T+24.891,R^{2}=0.92,C30_{W} \end{array}\right.$$
(6)$$\Delta_{G*/\sin\delta }=\left\{\begin{array}{l} -0.4626T+42.694,R^{2}=0.95,A15_{D}\\ -0.4702T+46.508,R^{2}=0.76,A30_{D}\\ -0.5171T+43.036,R^{2}=0.93,B15_{D}\\ -0.3696T+35.064,R^{2}=0.81,B30_{D}\\ -0.4229T+33.405,R^{2}=0.87,C15_{D}\\ -0.3281T+30.029,R^{2}=0.86,C30_{D} \end{array}\right.$$

From Equations (5) and (6), it can be seen that the Δ_G*/sinδ_ of the reclaimed asphalt is always linearly related to the test temperature, independent of the source and admixture of aged asphalt, and will not change after short-term aging.

### 3.4. Asphalt Fatigue Factor Evaluation Test Results and Analysis

An increase in rutting factor enhances the ability of the asphalt to resist permanent deformation under high-temperature conditions; however, a high rutting factor can lead to the asphalt being susceptible to cracking under low- and medium-temperature conditions. Therefore, the fatigue factor G*sinδ was introduced to characterize the ability of the asphalt to resist fatigue cracking under medium-temperature conditions after long-term aging. The variation curves of G*sinδ values with temperature for virgin and reclaimed asphalts after long-term aging are shown in Figure 7.

As can be seen from Figure 7, the G*sinδ gradually decreases as the temperature increases, and the G*sinδ of the reclaimed asphalt is larger than that of the virgin asphalt. The higher the amount of aged asphalt admixture, the larger the G*sinδ. The results of the nonlinear regression analysis showed that the G*sinδ was exponentially related to the temperature, and the correlation coefficients were all above 0.95.

Here, Δ_G*sinδ_ was selected as the dimensionless fatigue factor indicator for the reclaimed asphalt, and the effect of the relative fatigue factor of the reclaimed asphalt on its resistance to fatigue cracking under medium-temperature conditions was evaluated.

The variance results for the reclaimed asphalt Δ_G*sinδ_ and G*sinδ are shown in Table 6.

As can be seen from Table 6, the effect of the aged asphalt admixture on the fatigue resistance of reclaimed asphalt cannot be evaluated using G*sinδ as an indicator. The three *p*-values of Δ_G*sinδ_ are less than 0.05, indicating that the effects of the source and admixture on the rheological properties of the reclaimed asphalt can be assessed using Δ_G*sinδ_ as an indicator. Therefore, Δ_G*sinδ_ is suitable for characterizing the mid-temperature rheology of aged asphalt. The variation in Δ_G*sinδ_ of the reclaimed asphalt versus temperature is shown in Figure 8.

It can be seen that the Δ_G*sinδ_ of the reclaimed asphalt increases gradually with the increase in temperature. Even if the source of the aged asphalt is the same, the fatigue resistance of the reclaimed asphalt will vary with every 1% increase in admixture. The different reclaimed asphalts at different temperatures produced similar test results, as described above, which were similar to the relative viscosity and relative rutting factor test results.

The regression analysis showed that the linear relationship between the Δ_G*sinδ_ and temperature T is correlated well, and their regression equations are both Δ_G*sinδ_ = a*T* + b (a and b are fitting parameters), the calculation equation for which is shown in (7):(7)$$\Delta_{G*\sin\delta }=\left\{\begin{array}{l} 0.1657T+6.8668,R^{2}=0.67,A15_{C}\\ 0.823T-4.9943,R^{2}=0.88,A30_{C}\\ 0.1941T+4.1691,R^{2}=0.85,B15_{C}\\ 0.7204T-5.2205,R^{2}=0.89,B30_{C}\\ 0.1282T+4.2875,R^{2}=0.85,C15_{C}\\ 0.4313T-0.9126,R^{2}=0.83,C30_{C} \end{array}\right.$$

From Equation (7), it can be seen that the effects of the source and admixture of the aged asphalt on the fatigue resistance of the reclaimed asphalt can be similarly expressed by the slope and intercept in the linear relationship equation.

## 4. Conclusions

The addition of the aged asphalt increases the viscosity, rutting factor, and fatigue factor of the reclaimed asphalt, indicating that the high-temperature stability of the reclaimed asphalt is enhanced but the fatigue cracking resistance is attenuated.

The effect of the aged asphalt on the viscosity of the reclaimed asphalt can be evaluated using Δ*_η_*. The variance results showed that the value of Δ*_η_* depends on the source and the admixture of aged asphalt. At high temperatures, Δ*_η_* decreases linearly with increasing temperature, and its linear trend is independent of the source, the admixture, and whether it has been aged or not.

The effect of the aged asphalt on the viscoelasticity of reclaimed asphalt can be evaluated using Δ_G*/sinδ_ and Δ_G*sinδ_. The variance results showed that Δ_G*/sinδ_ and Δ_G*sinδ_ depend on the source and admixture of aged asphalt. At high temperatures, Δ_G*/sinδ_ decreases linearly with increasing temperature, and at medium temperatures, Δ_G*sinδ_ increases linearly with increasing temperature; both linear trends are independent of the source and admixture of aged asphalt.

## Figures and Tables

**Figure 1 polymers-14-03623-f001:**
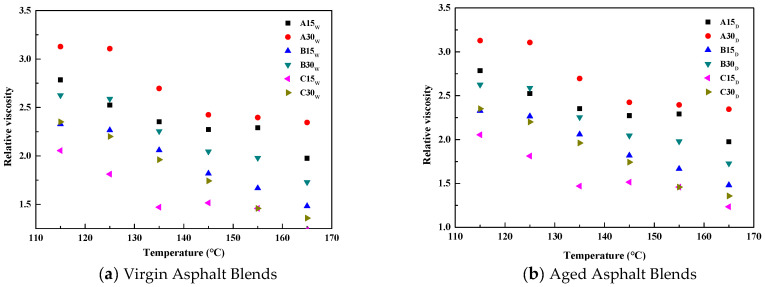
The results for the relative viscosity Δ*_η_* versus temperature.

**Figure 2 polymers-14-03623-f002:**
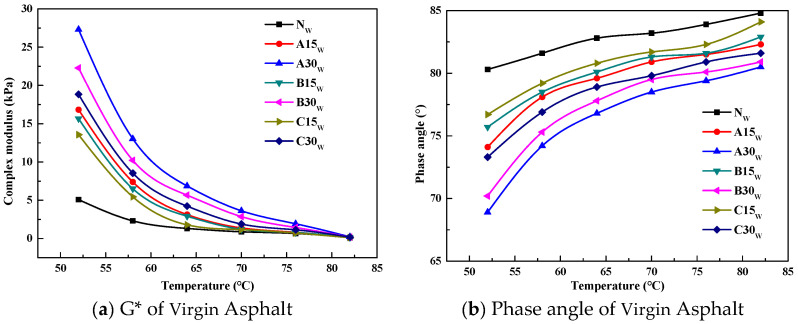
The results of the complex modulus and phase angle versus temperature for the unaged asphalt.

**Figure 3 polymers-14-03623-f003:**
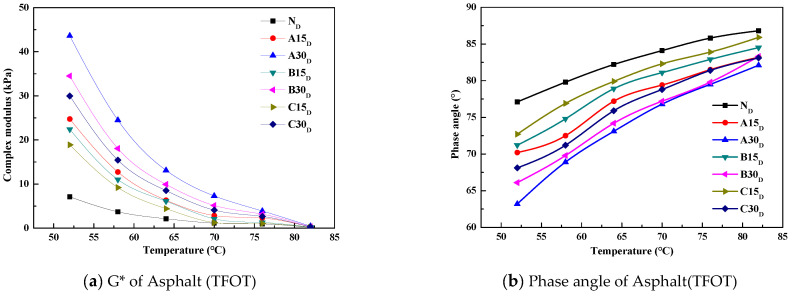
The results of the complex modulus and phase angle versus temperature for the short-term aged asphalt.

**Figure 4 polymers-14-03623-f004:**
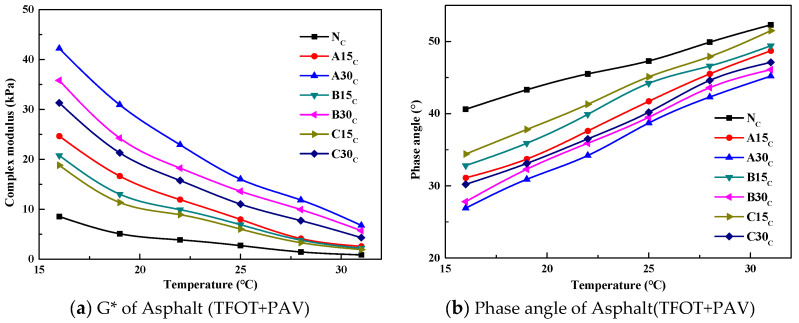
The results of the complex modulus and phase angle versus temperature for the long-term aged asphalt.

**Figure 5 polymers-14-03623-f005:**
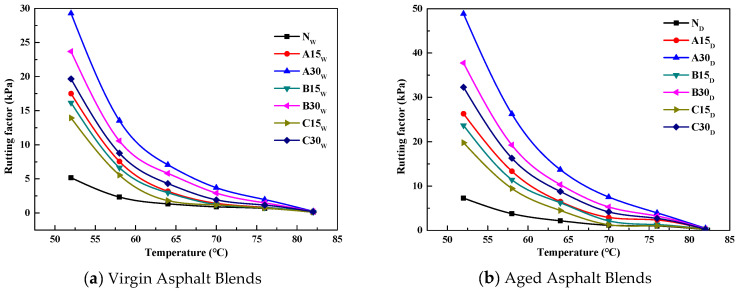
The results of the rutting factor versus temperature for unaged (**a**) and short-term aged (**b**) asphalts.

**Figure 6 polymers-14-03623-f006:**
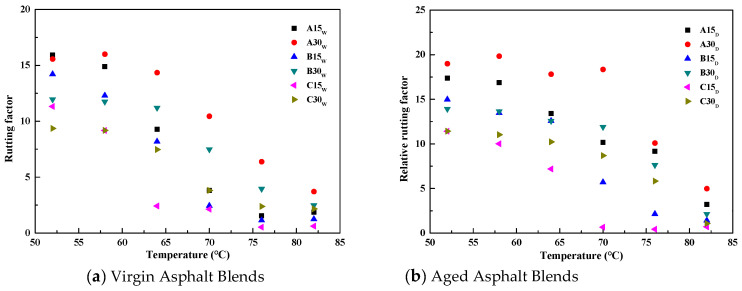
Variations of Δ_G*/sinδ_ for reclaimed asphalt versus temperature.

**Figure 7 polymers-14-03623-f007:**
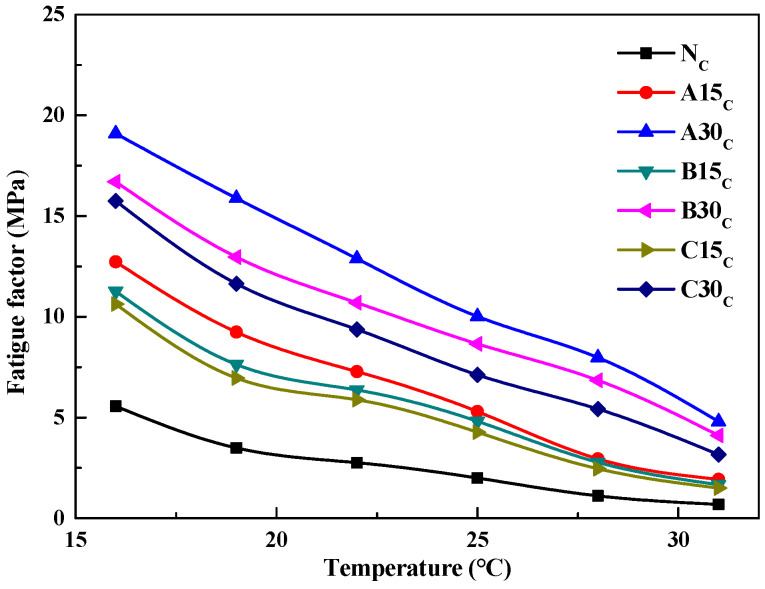
Variation of Δ_G*sinδ_ for asphalt versus temperature.

**Figure 8 polymers-14-03623-f008:**
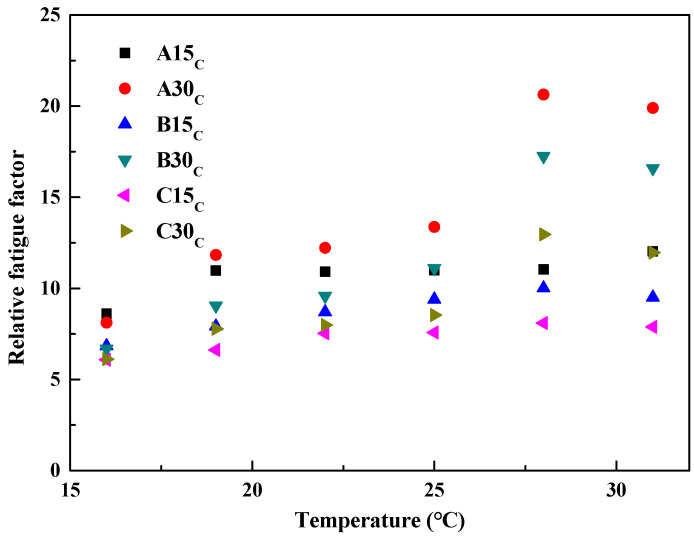
Variation of Δ_G*sinδ_ for reclaimed asphalt versus temperature.

**Table 1 polymers-14-03623-t001:** The technical indicators of aged asphalt.

Category	Penetration (25 °C/0.1 mm)	Softening Point (°C)	Ductility (15 °C/cm)	Viscosity (135 °C/Pa·s)
Test method	T0604	T0606	T0605	T0625
A	27	73	3.2	2.99
B	31	71	4.6	2.67
C	35	67	6.5	2.23

**Table 2 polymers-14-03623-t002:** The technical indicators of virgin asphalt.

Indicators		Test Results	Specification
Penetration (25 °C/0.1 mm)		87	80~100
Softening point (°C)		46.5	≥44
Ductility (15 °C/cm)		>100	≥100
Viscosity (135 °C/Pa·s)		0.36	-
Residue after TFOT	Mass loss/%	−0.48	≤±0.8
	penetration ratio (25 °C/%)	63.5	≥57
	Ductility (10 °C/cm)	12.9	≥8

**Table 3 polymers-14-03623-t003:** Viscosity results of virgin and reclaimed asphalt binders.

Asphalt Types	Test Temperature (°C)					
	115	125	135	145	155	165
N_W_, N_D_	0.93, 1.46	0.66, 1.03	0.38, 0.68	0.29, 0.44	0.23, 0.32	0.19, 0.27
A15_W_, A30_W_	1.33, 1.87	0.94, 1.29	0.52, 0.72	0.39, 0.53	0.31, 0.42	0.25, 0.33
B15_W_, B30_W_	1.29, 1.72	0.89, 1.19	0.56, 0.68	0.38, 0.52	0.30, 0.39	0.24, 0.31
C15_W_, C30_W_	1.23, 1.61	0.85, 1.12	0.48, 0.65	0.36, 0.45	0.28, 0.34	0.23, 0.28
A15_D_, A30_D_	2.07, 2.83	1.42, 1.99	0.92, 1.23	0.59, 0.76	0.43, 0.55	0.35, 0.46
B15_D_, B30_D_	1.97, 2.61	1.38, 1.83	0.89, 1.14	0.56, 0.71	0.40, 0.51	0.33, 0.41
C15_D_, C30_D_	1.91, 2.49	1.31, 1.71	0.83, 1.08	0.54, 0.67	0.39, 0.46	0.32, 0.38

**Table 4 polymers-14-03623-t004:** Analysis of variance results for reclaimed asphalt with Δ*_η_* and *η*.

Category	Sources of Aged Asphalt	Admixture of Aged Asphalt	Test Temperature	Aging Conditions
Statistical probability *p*-value (Δ*_η_*)	3.9 × 10^−5^	0.004	3.3 × 10^−7^	0.039
Statistical probability *p*-value (*η*)	0.851	0.100	2.9 × 10^−4^	0.013

**Table 5 polymers-14-03623-t005:** Analysis of variance results for reclaimed asphalt Δ_G*/sinδ_ and G*/sinδ.

Category	Sources of Aged Asphalt	Admixture of Aged Asphalt	Test Temperature	Aging Conditions
Statistical probability *p*-value (Δ_G*/sinδ_)	0.001	0.029	5.10 × 10^−7^	0.031
Statistical probability *p*-value (G*/sinδ)	0.533	0.056	1.35 × 10^−5^	0.068

**Table 6 polymers-14-03623-t006:** Analysis of variance results for reclaimed asphalt Δ_G*sinδ_ and G*sinδ.

Category	Sources of Aged Asphalt	Admixture of Aged Asphalt	Test Temperature
Statistical probability *p*-value (Δ_G*sinδ_)	0.009	0.016	0.009
Statistical probability *p*-value (G*sinδ)	0.524	0.003	2.87 × 10^−3^

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
