# Peer review of "Evaluation of the Rheological Properties of Virgin and Aged Asphalt Blends"

_polymers, 2022, doi:10.3390/polym14173623_

Round 1

Reviewer 1 Report

Attractive manuscript related to the assessment of the influence of aged asphalt on the rheological properties of reclaimed asphalt binder.

In general, the paper is well presented, but some details must be clarified or corrected/completed:

1.      Line 32: you wrote “Studies …”, but only one reference (from 2006) was cited ([7]);

2.      Line 35: please bear in mind that, for example, in many European countries, RAP material can only be stored in waterproofed locations (so the above-mentioned toxic substances leaching is not a problem);

3.      Table 1: please include the standards used in each test method;

4.      Table 2: where can we find these “Specifications”?

5.      Sections 2.2 and 2.3: please include the standards used;

6.      Line 80: “…production and construction, …” or “…production and application, …”;

7.      Line 88: what “AASHTO regulations” were used?

8.      Section “3. Results and discussion”: where is “discussion” with other results (from other researchers)? Perhaps this section deserves to be better developed, including further discussion with more inferences cited elsewhere (where appropriate);

9.      Line 93: this text must be placed before Table 3;

10.   Line 152: this text must be placed before Figure 2;

11.   Line 179: do you think that the performance of this “recycled” asphalt may be well assessed (at high temperatures) by the “rutting factor” (G*/sin δ)?

12.   Lines 235/236: this text must be placed before Figure 7;

13.   Lines 253: this text must be placed before Figure 8;

14.   Please also include (where appropriate) the relevant limitations of this study. Some uncertainties should also be pointed out (e.g., related to the statistical analysis);

15.   Some final sections have been omitted, namely: Author Contributions; Conflicts of Interest, etc.

Reviewer 2 Report

The studies on the influence of aged asphalt on the rheological properties of bituminous binder is interesting from scientific point of view.

11.Item 2 should be written as "Materials and method" instead of "Methodology".

22. Were the bitumen viscosity tests (point 2.2) carried out with a Brookfield viscometer?

33. It should be clarified if long-term aging was performed as TFOT + PAV?

44. On the basis of what standards was the aging performed by means of TFOT and PAV methods?

55. Item 3.2 should be written as "Asphalt viscoelasticity evaluation test results and analysis" instead of "Asphalt viscosity evaluation test results and analysis".

Reviewer 3 Report

Dear authors, 

thank you for your paper. I have provided several comments in the attached file. Additionally I have few questions.

1) were the three selected RAP materials somehow specific or you took them randomly? why only 3?

2) for the blending of aged binder and virgin what is the reason that you focused only on 15% and 30%? Is it because in your country regularly not more than 30% RAP is used is asphalt mixture?

3) what was the reason to using rutting parameter instead of MSCR test which is more popular for bitumen rutting behaviour?

Author Response

Dear Reviewers:

We are very grateful to Reviewer for reviewing the paper so carefully. We have tried our best to improve and made some changes in the manuscript.

Responds to the reviewers' comments:

1)were the three selected RAP materials somehow specific or you took them randomly? why only 3?

Response:In this study, aged asphalt was recovered from three different sources of RAP. The applicability of this conclusion to other aged asphalt will be verified later.

2) for the blending of aged binder and virgin what is the reason that you focused only on 15% and 30%? Is it because in your country regularly not more than 30% RAP is used is asphalt mixture?

Response:In our country, long-term road performance is considered. Generally, the proportion of recycled materials added in large-scale engineering applications does not exceed 30%

3) what was the reason to using rutting parameter instead of MSCR test which is more popular for bitumen rutting behaviour?

Response:Thank you very much for your valuable comments. In this study, the rutting factor was used to evaluate the performance of recycled asphalt, and the MSCR test could be used in the next study.

Round 2

Reviewer 3 Report

Dear authors, in the first review some of the comments were not addressed. Even a short explanation is perfect.

1) I mentioned some comment to the paper title. Can you respond, please? I agree that aged asphalt was received form reclaimed, but then it is not logic that in title you are writing about influence of aged asphalt on reclaimed asphalt since you are blending the aged with virgin and studying the blend.

2) I asked about short description of the Absem method since many readers will neither know what this method is about nor be able to read some reference to a standard.

3)  The different ductility temperatures: I understand that this comes from standards, but my question is why in general for virgin and TFOT aged binders different temperature is used. There must be some reason even behind the standards. I am interested why it was decided to put these different temperatures in the standard.

4) DSR test geometry: so only PP25 was used. Often for lower temperatures and especially aged binders PP8 is used as well. But clear to me. Just it is better to write it in the text as test geometry PP25, then everyone knows what is meant.

5) I would include the information about the 3.5% error for viscosity in the text. 

6) the sentence in the paragraph below figure 4 where you modified the wording because of Newtonian fluid, please, read the modified sentence again, since now it seems a bit incomplete. In the sentence something is missed.

7) I would include the provided explanation for high temperature stability in the text of the paper. Then it is crystal clear to all readers.
